# A Deep Learning-Enhanced Stereo Matching Method and Its Application to Bin Picking Problems Involving Tiny Cubic Workpieces

**Masaru Yoshizawa, Kazuhiro Motegi and Yoichi Shiraishi ***

Graduate School of Science and Technology, Gunma University, Ohta 373-0057, Japan;
t201b096@gunma-u.ac.jp (M.Y.); motegi@gunma-u.ac.jp (K.M.)
* Correspondence: yoichi.siraisi@gunma-u.ac.jp; Tel.: +81-276-50-2532

**Abstract:** This paper proposes a stereo matching method enhanced by object detection and instance segmentation results obtained through the use of a deep convolutional neural network. Then, this method is applied to generate a picking plan to solve bin picking problems, that is, to automatically pick up objects with random poses in a stack using a robotic arm. The system configuration and bin picking process flow are suggested using the proposed method, and it is applied to bin picking problems, especially those involving tiny cubic workpieces. The picking plan is generated by applying the Harris corner detection algorithm to the point cloud in the generated three-dimensional map. In the experiments, two kinds of stacks consisting of cubic workpieces with an edge length of 10 mm or 5 mm are tested for bin picking. In the first bin picking problem, all workpieces are successfully picked up, whereas in the second, the depths of the workpieces are obtained, but the instance segmentation process is not completed. In future work, not only cubic workpieces but also other arbitrarily shaped workpieces should be recognized in various types of bin picking problems.

**Keywords:** convolutional neural network; stereo matching; depth estimation; object detection; segmentation

## 1. Introduction

It is common to use deep convolutional neural networks (D-CNNs) [1] for image recognition to examine the appearance of products or for the autonomous driving of automobiles and mobile robots. This paper deals with the application of a D-CNN for solving bin picking problems consisting of tiny cubic workpieces. A bin picking problem is defined as a problem involving automatically picking up objects with random poses in a stack using a robotic arm. A tiny cubic workpiece is defined as a cube whose edge length is less than or equal to 10 mm. In the detection of defects on the surface of a product, D-CNNs can recognize an object less than 5 mm in length, as shown in a previous experiment [2]. However, this is a two-dimensional case, and the recognition of objects in three-dimensional space, such as during bin picking problems, requires depth estimation. To the best of the authors' knowledge, a bin picking method for picking up tiny workpieces, that is, a method with efficient depth estimation, has not been reported.

The cost performance of depth cameras or three-dimensional (3-D) scanners has been greatly improved in recent years, and these devices are experimentally used for solving bin picking problems. For example, a depth camera with an infrared sensor [3] is cost-effective, but noise reduction is necessary when it is applied to bin picking problems. Nishi et al. [4] improved the performance of such a camera and used it to solve a bin picking problem. A set of industrial pipes was their target for recognition, and the size of the pipes was larger than that of our workpieces. Khalid et al. applied a segmentation algorithm to images obtained using an RGB-D camera for estimating the depth of an object accurately [5]. In their study, the object was a euro-palette whose size was minimally 150 mm × 10 mm.

Sakata et al. [6] suggested a combination of object detection and segmentation algorithms for solving a bin picking problem. Their workpiece was a standard-sized smart phone, and only the downward movement of the robotic arm was controlled. Matsumoto et al. [7] used an RGB-D camera for recognizing household goods by applying a segmentation algorithm with D-CNN. In their study, each object's volume was lower or higher than 400 cm$^3$, which was larger than that of our workpieces. Therefore, the application of an RGB-D camera with an infrared sensor is limited thus far to bin picking problems with larger workpieces than those dealt with in this paper.

Three-dimensional scanners have been used for the recognition of objects in bin picking problems. Several studies reported the application of 3-D scanners; however, the size of their workpieces was larger than that dealt with in our study. Some papers clearly describe the size of the workpieces the authors used in the bin picking problem. Ly et al. [8] suggested a neural network that uses full and some partial clouds for training, and they applied it to the estimation of the grasping pose of objects. In their study, the size of each object was around 80 mm. Madhusudanan et al. [9] proposed a calibration method for an automated eye-in-hand robot-3D scanner to reduce stitching errors. In their study, a ball-bar whose size was 38 mm × 100 mm was the target for grasping. Xu et al. [10] proposed a method for training CNNs to predict instance segmentation results using synthetic data. They replaced the original feature extractor in the point clouds with dynamic graph convolutional neural networks, and the method was applied to the instance segmentation problem consisting of objects whose sizes were 58, 93, and 120 mm. Buchholz et al. [11] used a 3-D scanner to solve a bin picking problem and proposed a grasp planning method. They used two kinds of workpieces in the experiment, including 100 mm metal joint hangers and 200 mm piston rods.

Recently, stereo matching has been widely used in combination with hyperspectral imaging, which collects and processes information across the electromagnetic spectrum [12]. The electromagnetic spectrum is the range of frequencies (the spectrum) of electromagnetic radiation and their respective wavelengths and photon energies [13]. The objective of this approach is to find objects, identify materials, or detect processes by using an image corresponding to a specific wavelength. Two kinds of research fields may relate to the problem dealt with in this paper. The first research field is anomaly detection [14]. Sheng Lin et al. [15,16] proposed a combination of dual collaborative constraints corresponding to the background and anomaly components by applying a low-rank and sparse representation technique model. They verified the efficiency of the proposed method against four datasets, and the images were images of urban areas with a resolution ranging from 1 m to 3.7 m. The second research field is object detection via stereo matching with high-resolution images obtained using hyperspectral cameras, as depicted in [17]. In that study, two mast cameras with different wavelength ranges and different focal lengths were mounted on a Mars Rover and used for the fusion of the obtained images and their alignment. Moreover, the suggested pansharpening algorithm generates high-resolution images. This algorithm may be useful for our bin picking problem; however, the depth of objects was not dealt with in that study, and the objects were environmental scenes around the Mars Rover.

As mentioned above, the sizes of objects for successful bin picking using an RGB-D camera or a 3-D scanner are much larger than those of the objects dealt with in this study. In the trials involving 3-D imaging using an RGB-D camera or a 3-D scanner to pick up tiny cubic workpieces, as described in Section 2, the obtained images are too vague to be used for image recognition. Such hyperspectral imaging approaches for the separation of objects from the background in an image, enhancement of image resolution, or object detection using images with abundant information may be useful for our bin picking problem; however, object detection or segmentation of tiny objects, which is the problem of interest in this paper, has not yet been dealt with. Moreover, at present, hyperspectral cameras are around 40 times more expensive than RGB cameras, and the cost problem should be considered in industrial applications. To solve the bin picking problem involving tiny cubic workpieces, this paper suggests a method combining instance segmentation

and stereo matching [18,19] with a pair of RGB cameras. The basic idea is to obtain a highly precise 3-D map describing a stack consisting of tiny cubic workpieces by using the proposed method. The novelty of this paper is the use of a D-CNN to enhance the conventional stereo matching method and the application of the suggested method to a bin picking problem consisting of tiny cubic workpieces that are 10 mm or 5 mm cubes.

The structure of this paper is as follows: In Section 2, the stereo matching problem is defined, and the images obtained using an RGB-D camera or a 3-D scanner for the problem are shown. The suggested enhancement method is explained in Section 3. Section 4 provides a description of the experimental results for the recognition of bins and the picking processes using a robotic arm. The objective of this study is to verify the feasibility of the suggested method. Therefore, the strengths and weaknesses are discussed in Section 5. Finally, the conclusions and directions for future work are presented in Section 6.

## 2. Bin Picking Problem Consisting of Tiny Workpieces

Bin picking problems are quite common and important in product manufacturing. This paper deals with a bin picking problem consisting of tiny cubic workpieces. In product manufacturing, the dimensions, material, and weight of objects are the same, and the bin picking algorithm dealt with in this study has a wide variety of applications involving picking up objects using a small robotic arm. For example, the technology of picking up a small chip using a robotic arm, as shown in Figure 1, is demonstrated in [20]; however, this is not a bin picking problem because all chips are pre-aligned on the stage.

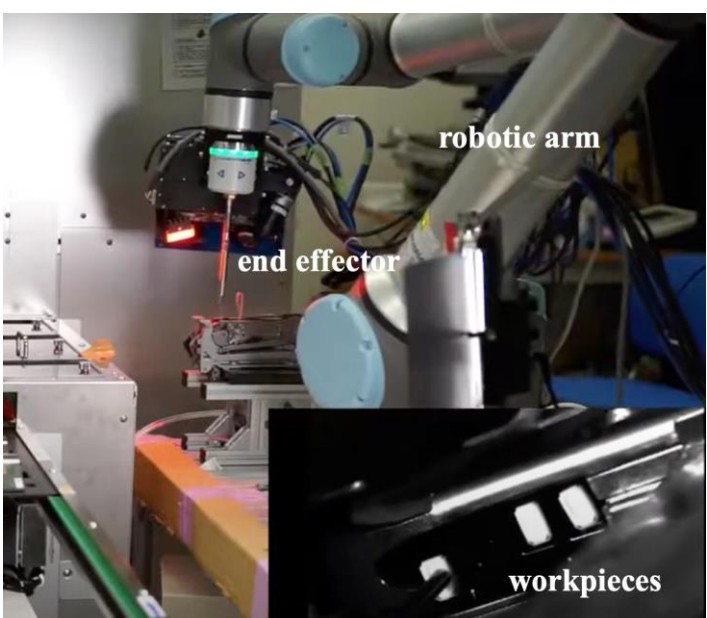

**Figure 1.** Example of a workpiece being mounted on a 3-D board using a robotic arm.

We fabricated a robotic hand that accurately reproduces the human hand skeleton with the corresponding muscles and tendons [21]. Figure 2 shows an example of the robotic hand picking a workpiece with a size of 20 mm × 30 mm, which is a material used for cutting out dentures. In the future, the work of inserting a 1 mm diameter lead wire into a connector is planned to be executed by the robotic hand. The bin picking problem, consisting of tiny workpieces, must be solved to automate this work. Here, the depth of each of the workpieces in the stack must be accurately estimated before the picking process.

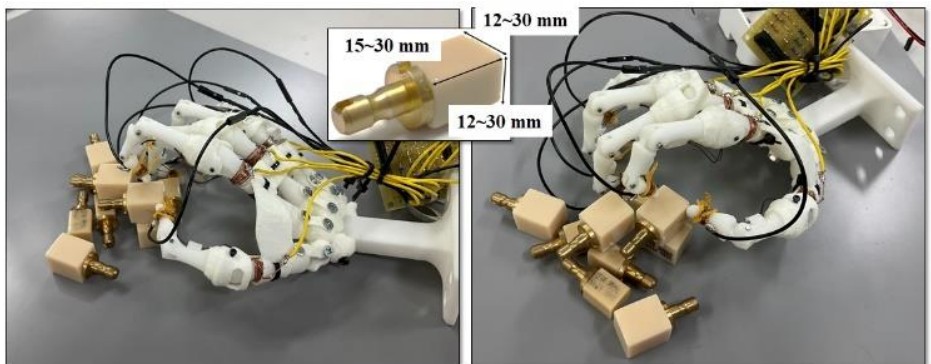

**Figure 2.** Future assembly work with tiny workpieces.

A depth camera, Intel Realsense D435i [3], was applied to take the image of a stack consisting of tiny workpieces, and the obtained image is shown in Figure 3. The distance between the camera and the stack was set to 20 cm, which is the minimum distance in the specification because the exact shape of the workpieces needs to be recognized. As shown in Figure 3, it seems difficult to visually recognize the stack in both the upper and side images. Next, a 3-D scanner, Artec Eva [22], was used for scanning the same stack, and the obtained image is shown in Figure 4. In this figure, the contour of the workpieces is deformed, and it might be difficult to recognize the stacking of the workpieces. The workpieces are cubes whose edge length is 10 mm or 5 mm and are made of resin and rubber, respectively. Figure 5 shows examples of bulk stacks and workpieces. Each bulk stack has an arbitrary number of workpieces and an arbitrary number of layers. Generally, the workpieces used in the fabrication process have a uniform size and shape.

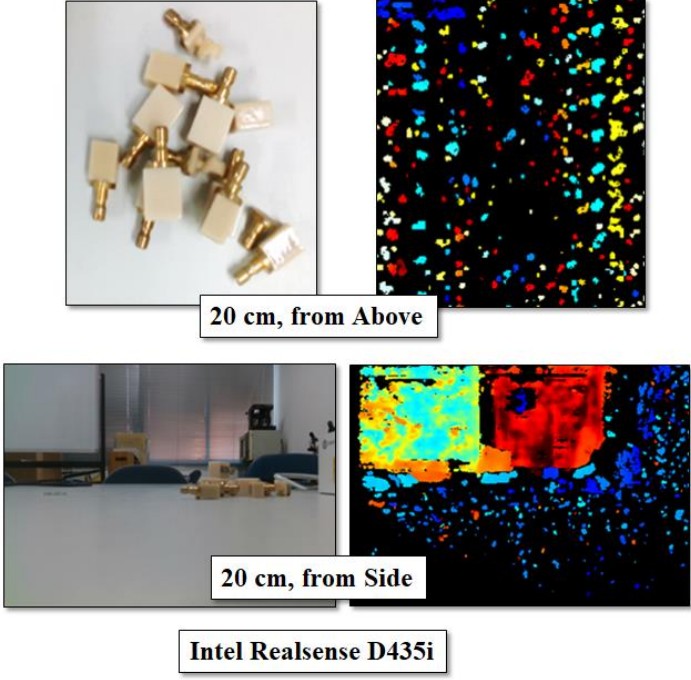

**Figure 3.** Images obtained using a depth camera.

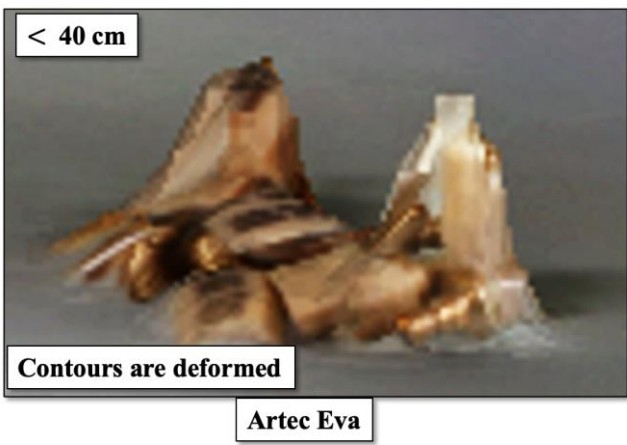

**Figure 4.** Image obtained using a 3D scanner.

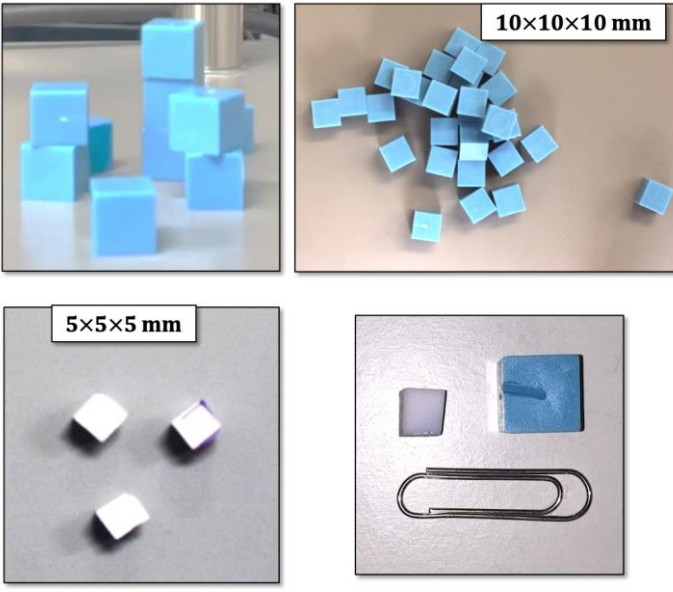

**Figure 5.** Bin picking problem and tiny workpieces.

## 3. Suggested Stereo Matching Method

The stereo matching method has already been suggested in previous research, and it estimates the distance (depth) between a camera and an object by using the parallax of two cameras arranged side by side with a specified distance [18,19]. When this method is applied to object observation, depth is given to each pixel of the surface, and it is not necessarily the same around the pixel (an example of depth calculation is shown below). Therefore, the obtained depth must be modified based on the outline of the object by using its CAD model. This modification is particularly necessary for tiny workpieces because the objects do not have enough pixels to estimate their depth. We suggest an enhanced stereo matching method by using a deep convolutional neural network (D-CNN) [23] and the Harris corner detection algorithm [24].

### 3.1. System Configuration

As shown in Figure 6, the suggested system consists of three subsystems. "Object Detection & Segmentation" generates a 2-D map from the image of an object. "Depth Estimation" and "Rotation Estimation" control the robotic arm. The former estimates depth, that is, the distance between the camera and the object, and the latter operates the end effector of the robotic arm. The process "Contact Estimation between Robotic Arm and Workspace" creates a picking plan for each layer of workpiece stacking by using the

contact of the end effector with the tiny workpieces. Finally, the robotic arm is controlled based on the plan.

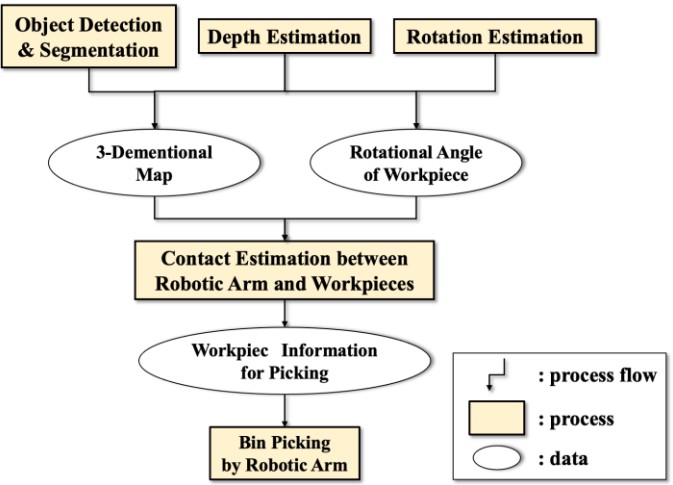

**Figure 6.** Suggested system configuration.

*3.2. Process Flow*

The process flow is shown in Figure 7. The stacked workpieces are layered in the z-axis direction based on their depths. Firstly, the image taken by the left camera is used for object detection and instance segmentation based on the D-CNN to obtain a 2-D map of stacking. Secondly, the depth of each workpiece is obtained by using the conventional stereo matching method. Finally, a 3-D map for bin picking is generated by synthesizing the 2-D map of the stack and the depth of each workpiece. The right side is the part responsible for controlling the picking movement of the robotic arm. The corners of all workpieces within the image are detected using the Harris corner detection algorithm (abbreviated as the Harris algorithm) [24], and the angle of each workpiece with the horizontal axis is calculated. One of the workpieces for pick-up is chosen in the 3-D map by checking the possibility of contacts between the end effector and other workpieces. Finally, the chosen workpiece is picked up by controlling the robotic arm, and the above processes are repeated for picking up all workpieces.

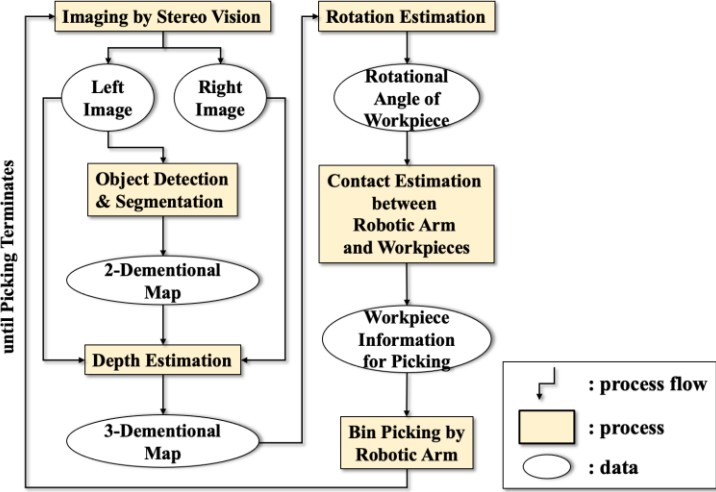

**Figure 7.** Process flow of the proposed system.

### 3.3. Object Detection and Segmentation

Mask R-CNN [23] is applied to both object detection and instance segmentation in the image taken by the left camera. As for object detection, the outputs of Mask R-CNN include the rectangle enclosing the detected object with a label and confidence, while the filled area is generated as the detected object.

### 3.4. Depth Estimation

The conventional stereo matching algorithm is applied for depth estimation of the tiny workpieces; however, the tiny workpieces cause an error problem. Depth is obtained for each pixel in the image, and the problem is that the depths of adjacent pixels are not necessarily the same, and there are pixels whose depth is not obtained. This problem is especially conspicuous for tiny objects. An example is the "Stereo Matching" image shown in Figure 8. It is necessary to obtain the contour of the object, and the novelty of this paper is its suggestion of combining object detection and instance segmentation based on D-CNN to solve this problem.

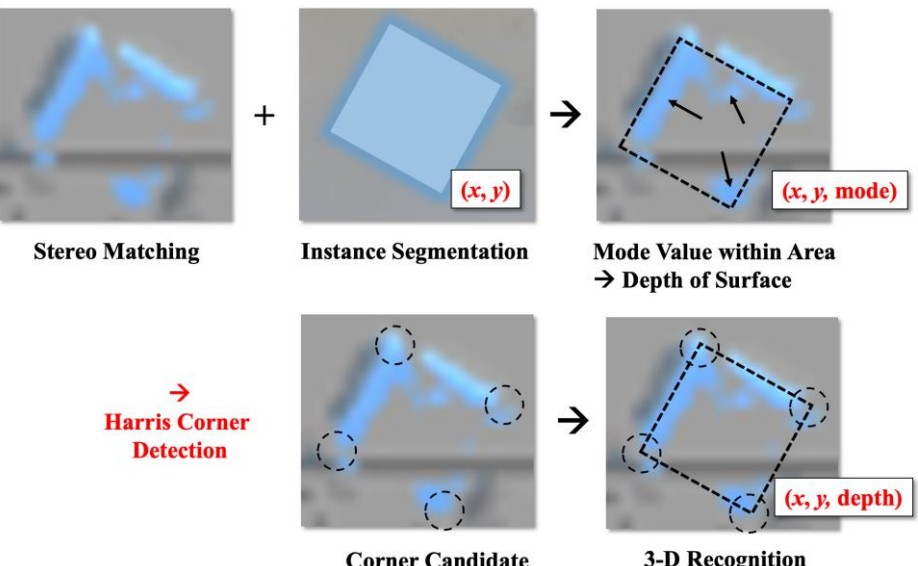

**Figure 8.** Synthesis of depth values and instance segmentation.

The process of recognizing objects in three-dimensional space by synthesizing the results of stereo matching and those of instance segmentation is shown in Figure 8. Here, there are some pixels whose depth could not be obtained, and they are shown as transparent pixels in the "Stereo Matching" image. Furthermore, there are cases where the adjacent pixels, which are estimated on the same surface, have different depths because instance segmentation is executed on pixels and has some errors. To solve these problems, the positions $(x, y)$ of pixels on an estimated surface obtained through instance segmentation and the depth corresponding to $(x, y)$ are first synthesized by matching their coordinates, and then the depths of pixels over the estimated surface are determined as $(x, y, depth)$, where *depth* is defined as their mode, except for undefined depths. The process of estimating the rotational angle of the workpieces utilizes the Harris algorithm, which detects the corner pixels of a given 3-D image. Here, the Harris algorithm also has some errors, and more than four points in the neighborhood of the true corner points are detected as candidate corners. Moreover, they are not necessarily in the area of the workpieces. As shown in Figure 5, this paper deals with cubic workpieces whose heights are the same, and we define the number of layers, $k$ (=1, 2, . . . , n), in the bulk stack. Therefore, if the depth differences among the corners are within half of the height of the workpieces, the layer number is accurately obtained. Its accuracy against actual bin picking problems is verified in the experiments by defining evaluation metrics.

### 3.5. Rotational Angle of Workpieces

The basic idea of calculating the rotational angle of each workpiece is to choose two points in the set of corner candidate points obtained using the Harris algorithm. The algorithm is shown below.

**[Rotational Angle Calculation Algorithm]**

(1)　Input the set of corner candidate points, $C = \{(c, y_{1s}), (x_{max}, y_{2s}), (x_{3s}, y_{min}), (x_{4s}, y_{max})\}$, obtained using the Harris algorithm, where, for example, "$(x_{min}, y_{1s})$" means all points having $x_{min}$, $x_{max}$, and $x_{min}$, $y_{max}$ are the minimum and maximum coordinates of $x$ and $y$ axes, respectively, in the set of corner candidate points;

(2)　Calculate the Euclidian distances of all pairs in $C$;

(3)　Sort all pairs in ascending order based on the distance calculated in (2);

(4)　Choose the top two corner candidate points included in the ordered list obtained in (3), as shown in Figure 9a;

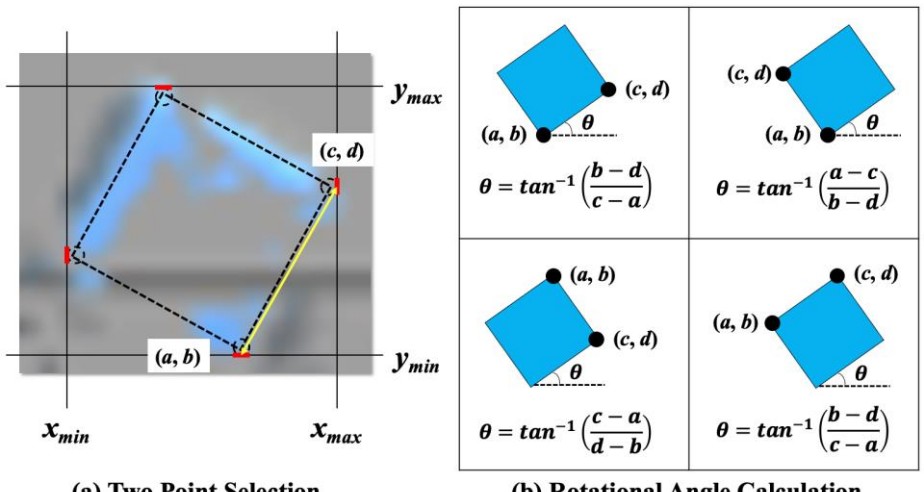

**(a) Two Point Selection**　　　　**(b) Rotational Angle Calculation**

**Figure 9.** Rotational angle calculation.

(5)　Calculate the rotational angle with the horizontal axis in the four cases, as shown in Figure 9b.

### 3.6. Contact Estimation of End Effector with Workpieces

When picking up a workpiece, it must be checked whether the end effector of the robotic arm may contact the workpieces in its neighborhood. The contact estimation is shown in Figure 10.

The operating 3-D space of the end effector is calculated, and the workpieces within the space are estimated for their possibility of being in contact with the end effector. As shown in Figure 10, in the top- and side-view coordinate systems, the system with red characters is the coordinate system with bulk stacking of the workpieces, and the system with black characters is the coordinate system within the robotic arm's operational space. The former coordinate system should be transformed into the latter one, which is the operating space of the robotic arm, that is, the height "$h$" and the range of angle "$\Theta$" in the cylindrical space with the diameter "$l$". The angle "$\Theta$" varies according to which face of the workpiece the robotic arm grasps. When other workpieces are included in the part of the cylindrical space determined by "$h$", "$l$", and "$\Theta$", this workpiece should not be picked up and another one is checked.

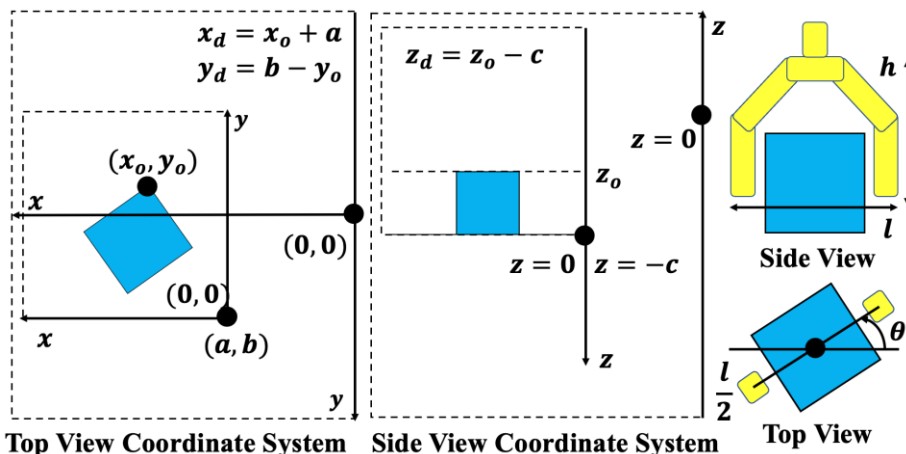

**Figure 10.** Contact estimation of the end effector with workpieces.

## 4. Experimental Results

The suggested deep learning-enhanced stereo matching method was applied to the bin picking problem, and its functions and performances were evaluated.

### 4.1. Experimental System

The experimental system's setup is shown in Figure 11. Two DFK33UP1300 cameras (ARGO CORPORATION, Osaka, Japan) [25] were used for the stereo vision, and their distance was 16.0 cm. The image size was 1280 × 1024 pixel, and DOBOT Magician [26], with its programming environment, was used as the robotic arm. The first bin picking problem is shown in Figure 12, where 10 plastic workpieces, whose edge size is 10 mm, are stacked into three layers.

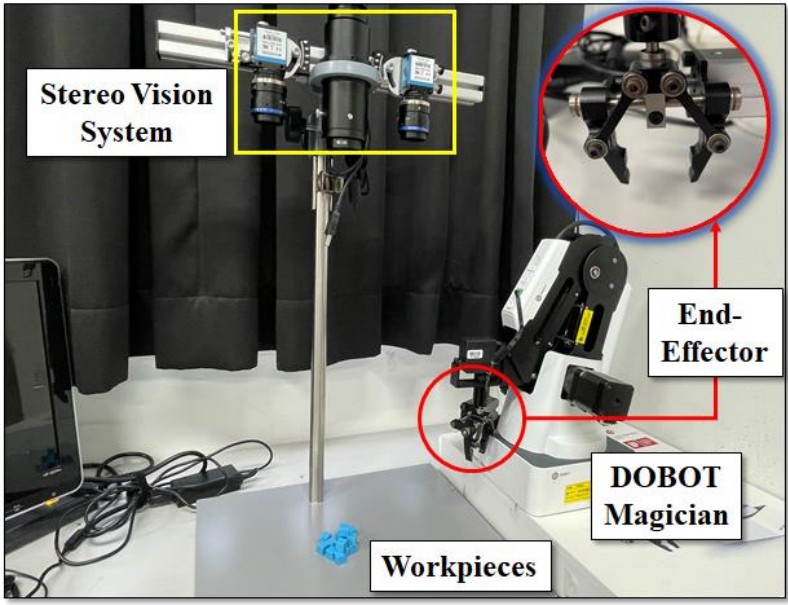

**Figure 11.** Experimental system.

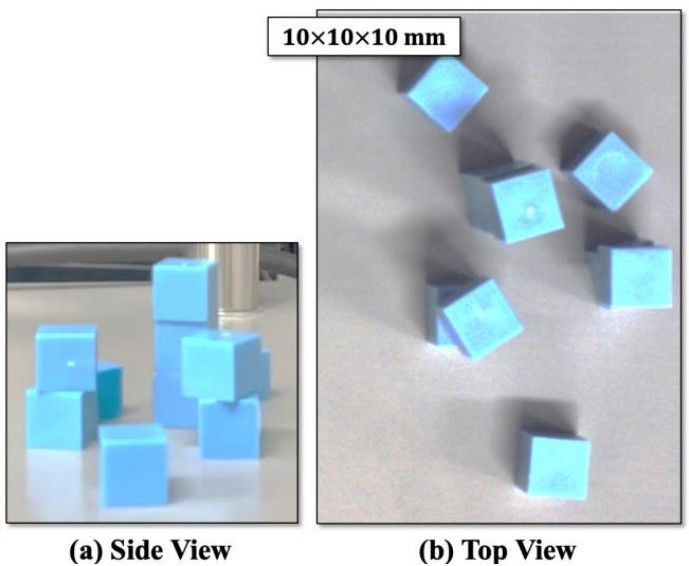

**(a) Side View**    **(b) Top View**

**Figure 12.** Bin picking problem for the experiment.

### 4.2. Evaluation Metrics of Depth Estimation

The most important function of the proposed system is depth estimation. Therefore, the evaluation metric of depth estimation is the accuracy of the estimated depth, approximated as the layer number in the stack, and it is calculated using the following expression. The 3-D size, that is, the length, width, and height of each workpiece, is uniform, and the estimated layer number is rounded down if it is less than half the height; otherwise, it is rounded up.

$$\text{Accuracy (A)} = \frac{\text{EstimatedLayerNumber (EL)}}{\text{ActualLayerNumber (AL)}} \times 100$$

### 4.3. Object Detection and Instance Segmentation

Mask R-CNN was used for object detection and instance segmentation processes. During training, the VGG Image Annotator [27] was used for annotation, and 60 images were used. An example of an image used in the training is shown in Figure 13. The positions of the workpieces in the stack, shown in Figure 12, were obtained via object detection with greater than 0.99 confidence, and the contour of the workpieces was extracted via instance segmentation, as shown in Figure 14a. The numbers from 0 to 5 in Figure 14b are instance numbers given to each of the recognized workpieces through instance segmentation.

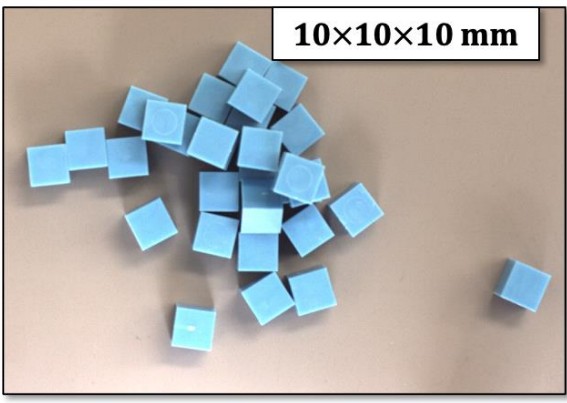

**Figure 13.** Image of a stack of workpieces used for training.

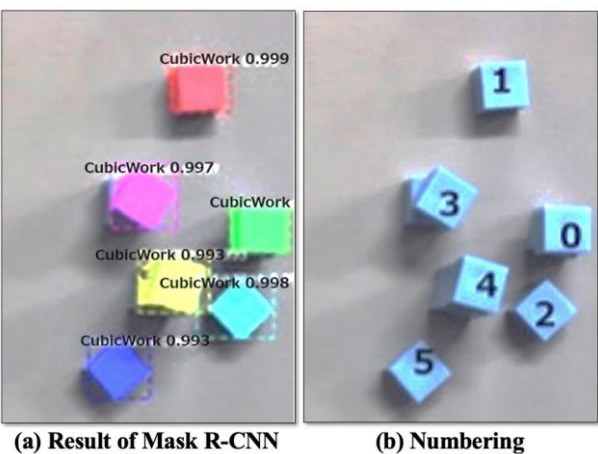

**(a) Result of Mask R-CNN**   **(b) Numbering**

**Figure 14.** Object detection and instance segmentation results for the bin picking problem.

### 4.4. Application of Enhanced Stereo Matching

The commercial software tool IC3D [28] was used for the conventional stereo matching process. The obtained result is shown in Figure 15; Figure 15a,b are the images obtained by the left and right cameras, respectively. Figure 15c is the stereo matching result generated by IC3D, and only some parts of the depths are obtained, as mentioned in Section 3.4.

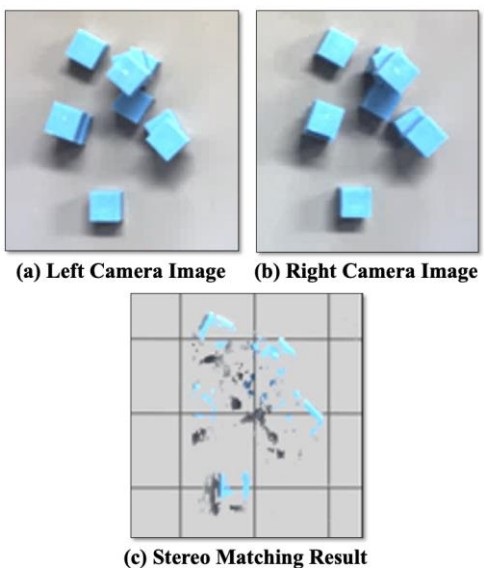

**(a) Left Camera Image**   **(b) Right Camera Image**

**(c) Stereo Matching Result**

**Figure 15.** Obtained result using conventional stereo matching.

As shown in this figure, the pixels with depth are shown in light blue; however, the visual determination of which pixel belongs to which workpiece seems difficult. This may be because the size of the workpieces is too small for the stereo matching software. This is the motivation for suggesting deep learning enhancement for stereo matching. For each workpiece, the pixel with depth and the pixel on the surface recognized through instance segmentation are matched by their $(x, y)$-coordinates, and the depth of the workpiece surface is determined assuming that the surface is flat. It seems that there are no pixels whose depth is not obtained, as there were no such cases when we applied the suggested method to the bin picking problem.

The next step is the application of the Harris algorithm, and we used the "cornerHarris" function [29] in the computer vision library in OpenCV. The parameters of block size and kernel size and the free parameters were set to 4, 3, and 0.04, respectively. The detected corner point candidates (set of pixels) are shown as red points in Figure 16. Figure 16a shows an example of a stack consisting of three layers, and Figure 16b shows the case of

one layer. In these cases, the corner points that were visually estimated are included. Some pixels are mistakenly recognized as corner points; however, these points can be excluded by checking their $(x, y)$ coordinates.

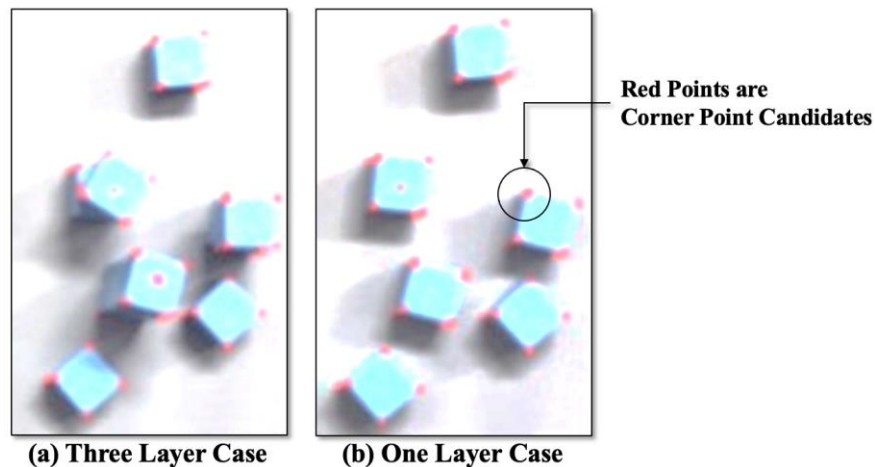

**(a) Three Layer Case**          **(b) One Layer Case**

**Figure 16.** Detected corner point candidates using the "cornerHarris" function.

### 4.5. Bin Picking by Robotic Arm

The "Bin Picking by Robotic Arm" process shown in Figure 6 creates a picking plan for the given bin picking problem. This plan is represented by a table, and the robotic arm picks one workpiece at a time based on the table. The picking plan generated for the bin picking problem shown in Figure 12 is presented in Table 1. As shown in the "Estimation of Layer Number (EL)", the accuracy of depth estimation is 100%.

**Table 1.** Bin picking plan.

| No. | Coordinates mm | | | Estimated Layer Number (EL) | Angle [deg] | Contacts |
|---|---|---|---|---|---|---|
| | *x* | *y* | *z* | | | |
| 0 | 222 | 47 | 22 | 2 | | no |
| 1 | 208 | 19 | 12 | 1 | | no |
| 2 | 218 | 69 | 12 | 1 | | no |
| 3 | 196 | 43 | 21 | 2 | | no |
| 4 | 202 | 61 | 31 | 3 | 83 | no |
| 5 | 190 | 82 | 10 | 1 | | no |

The picking plan consists of 3-D coordinates: mm, angle [deg], and contacts between the end effector and the workpieces. The origin in the $xy$-coordinate system is set to the top-left pixel, and the origin of the $z$-coordinate (depth) is on the stage above which the stack of the workpieces is placed. As shown in Figure 10, the angle is defined as the smaller angle between the end effector and the $x$-axis. When the end effector rotates clockwise to grip a workpiece and it contacts another workpiece "$w$", it is represented as "c:w" in the column "Contacts". Similarly, "cc:w" is filled when the rotation is counterclockwise. In the experiment, the height of the workpieces is 10 mm, and the number of layers is three. As shown in Table 1, the $z$-coordinates (depth) are 10, 12, 21, 22, and 31, corresponding to the layers in which the workpieces are placed. This means the maximum error of the estimated depth is 2 mm, and it is not a problem if the maximum error is less than half of the workpiece height in the given bin picking problem, as shown in the evaluation metrics. According to the picking plan shown in Table 1, workpiece "No. 4" is chosen in the first picking, as shown in Figure 17.

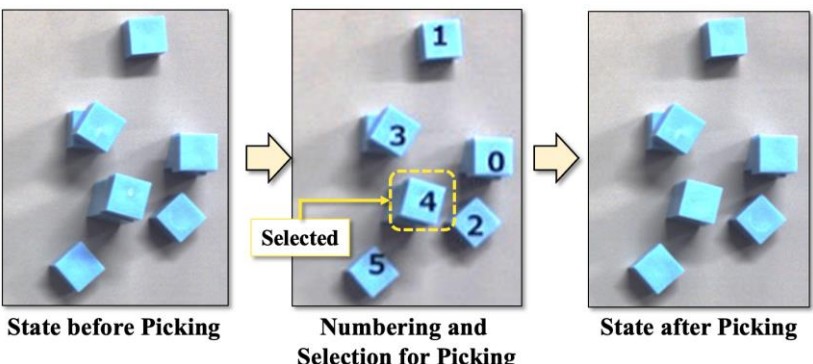

**Figure 17.** First picking selection.

By using the image after removing the "No. 4" workpiece, the stereo matching process is repeated. As a result, the picking plan shown in Table 2 is generated, and the picking process is executed based on the plan. Here, workpieces "No. 1", "No. 3", and "No. 5" are on the top layer, and it is shown that the end effector contacts with the workpieces "No. 3" and "No. 5" when they are picked. Based on this result, for workpiece "No. 3", a clockwise rotation is chosen for the end effector. Then, workpieces "No. 5" and "No. 1" are picked up in this order. The picking process is shown in Figure 18. The maximum error of depth estimation is 2 mm, and the accuracy of depth estimation is 100%.

**Table 2.** Bin picking plan for remaining workpieces.

| No. | Coordinates mm | | | Estimated Layer Number (EL) | Angle [deg] | Contacts |
|-----|-----|-----|-----|-----|-----|-----|
| | $x$ | $y$ | $z$ | | | |
| 0 | 208 | 19 | 13 | 1 | | no |
| 1 | 222 | 51 | 22 | 2 | 84 | no |
| 2 | 218 | 69 | 12 | 1 | | no |
| 3 | 195 | 44 | 21 | 2 | 54 | cc:5 |
| 4 | 190 | 83 | 10 | 1 | | no |
| 5 | 201 | 64 | 21 | 1 | 77 | cc:3 |

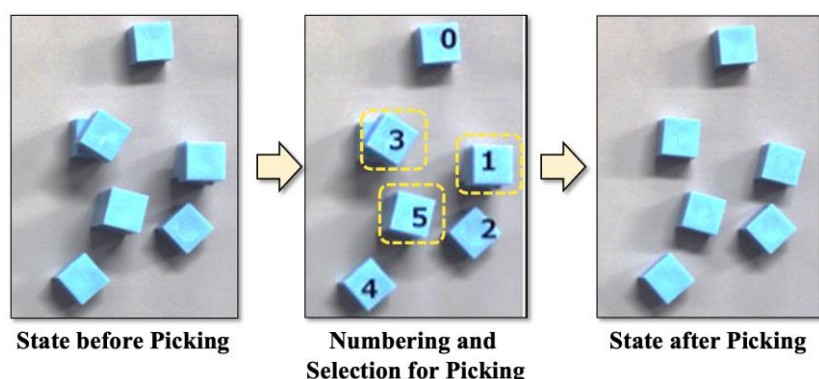

**Figure 18.** Second picking process.

All workpieces are similarly picked up, and thus, the given bin picking problem is solved. The next picking process is shown in Table 3 and Figure 19, and the maximum error of depth estimation is 2 mm. In this stage, the accuracy of depth estimation is 100%.

**Table 3.** Bin picking plan for the final stage.

| No. | Coordinates mm | | | Estimated Layer Number (EL) | Angle [deg] | Contacts |
|---|---|---|---|---|---|---|
| | *x* | *y* | *z* | | | |
| 0 | 193 | 41 | 11 | 1 | 25 | cc:5 |
| 1 | 208 | 19 | 12 | 1 | 14 | no |
| 2 | 221 | 48 | 12 | 1 | 73 | cc:1 |
| 3 | 218 | 69 | 12 | 1 | 42 | cc:0 |
| 4 | 190 | 82 | 10 | 1 | 43 | c:2 |
| 5 | 195 | 65 | 11 | 1 | 77 | c:4 or cc:0 |

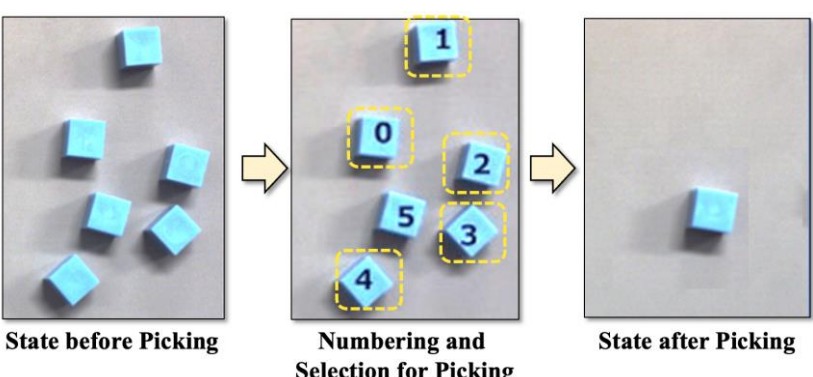

**State before Picking**     **Numbering and Selection for Picking**     **State after Picking**

**Figure 19.** Bin picking process for the final stage.

The picking operation of the workpieces by the robotic arm in the first stage is shown in Figure 20 as an example. At the end, the robotic arm successfully picks up all workpieces.

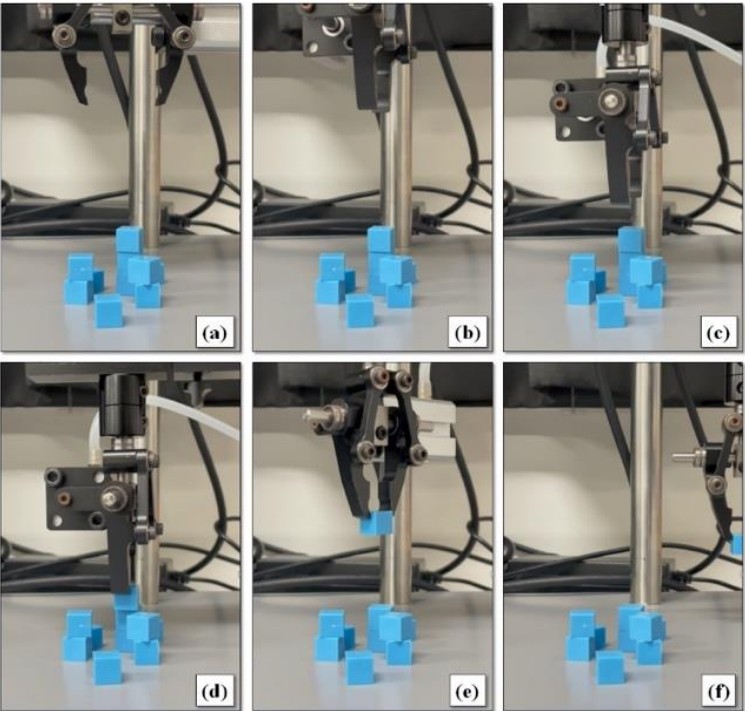

**Figure 20.** Example of picking by the robotic arm, and the first workpiece is picked up as shown in the processes from (**a**) to (**f**).

### 4.6. Bin Picking Problem with Much Tinier Workpieces

The application of the bin picking problem to much tinier workpieces is described in this section. This problem consists of five cubic workpieces whose edge length is 5 mm, and

they are stacked into three layers, as shown in Figure 21. The top is a slanted workpiece, and its size is not uniform because this workpiece is made from a rubber eraser by hand. This workpiece seems to be the smallest workpiece that could be successfully grasped by DOBOT Magician after repeating some trials.

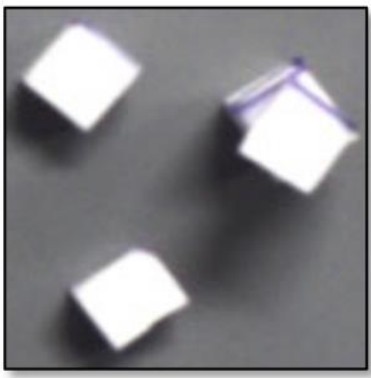 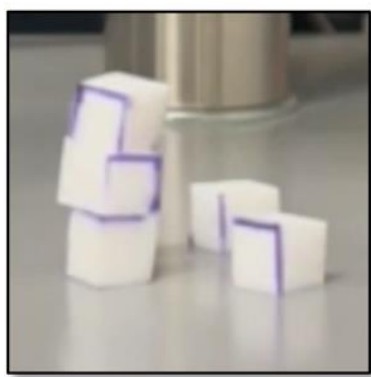

**Figure 21.** Example of a bin picking problem consisting of tinier workpieces.

The results of object detection and instance segmentation in each of the four stages are shown in Figure 22. The picking plan for the first, second, and third stages is shown in Table 4. Also, the accuracy of depth estimation is 100%. Except for the fourth stage, object detection and instance segmentation are successful in the other stages, with the confidence being equal to or greater than 0.94. The reason for the failure of object detection and instance segmentation in the fourth stage is likely that these workpieces are not included in the training images because the same workpiece placed in the upper left on the table, as shown in Figure 22c, is recognized with a confidence of 0.95. As shown in Figure 23, the above workpiece has a black line, and the side of another workpiece is dented because these workpieces are made by hand.

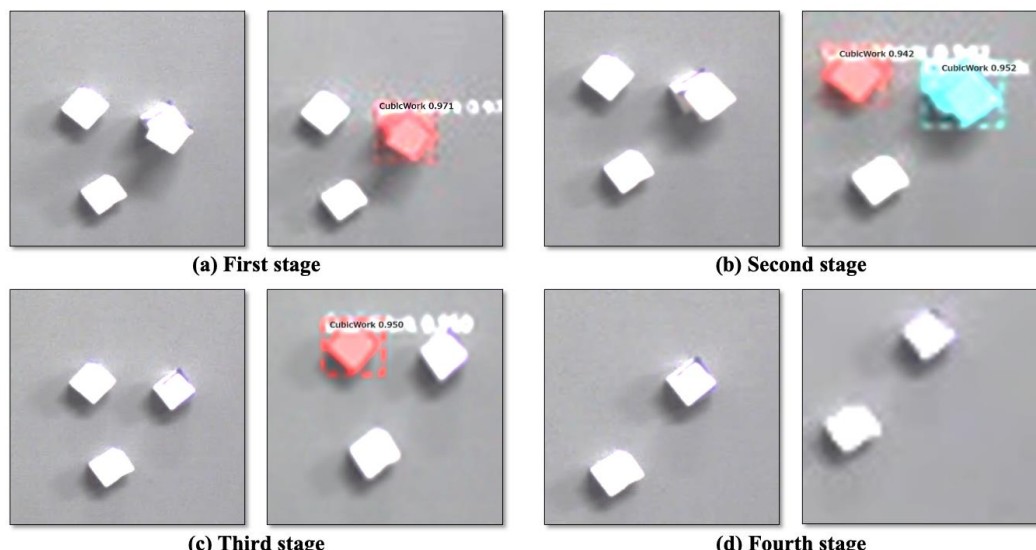

**Figure 22.** Input images and instance segmentation results in each stage.

**Table 4.** Bin picking plan for a problem consisting of tinier workpieces.

| No. | Coordinates mm | | | Estimated Layer Number (EL) | Angle [deg] | Contacts |
|---|---|---|---|---|---|---|
| | $x$ | $y$ | $z$ | | | |
| | First stage | | | | | |
| 0 | 196 | 100 | 16 | 3 | 41 | no |
| | Second stage | | | | | |
| 0 | 196 | 99 | 9 | 2 | 32 | no |
| 1 | 181 | 97 | 6 | 1 | 50 | no |
| | Third stage | | | | | |
| 0 | 181 | 97 | 6 | 1 | 50 | no |

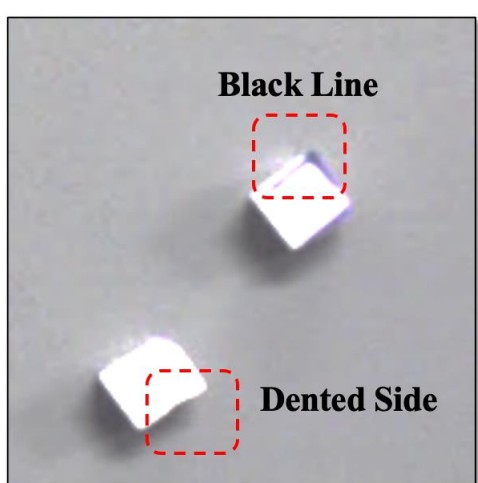

**Figure 23.** Workpieces with recognition failure.

As for depth estimation, the obtained $z$-coordinates are 6, 9, or 16 mm, corresponding to their true values of 5, 10, or 15 mm, respectively, and the errors are $\pm1$ mm, which are less than half of the workpiece height. In conclusion, the suggested depth estimation method is practical for tinier cubic workpieces whose edge lengths are 5 mm. The picking process by the robotic arm in the first stage is shown in Figure 24. As shown in the processes (a) to (c), the target workpiece is successfully picked up.

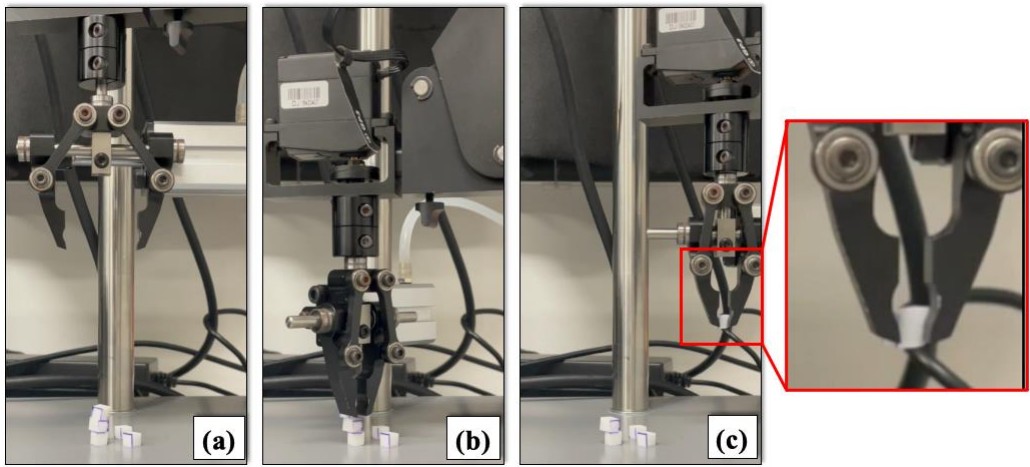

**Figure 24.** Picking process by the robotic arm in the first stage, and the target workpiece is picked up as shown in the processes from (**a**) to (**c**).

## 5. Discussion

The suggested deep learning-enhanced stereo matching method is efficient for depth estimation of tiny cubic workpieces, with the exception of one failure in the experiments.

This failure is likely to be solved by improving the quality and quantity of training images because this is a failure of object detection and segmentation. Moreover, the pickup of workpieces is successful during the actual operations of the robotic arm if the objects are properly recognized. These results show the novelty and effectiveness of the suggested method. On the other hand, verification of the versatility of the suggested algorithm is not sufficient, and other bin picking problems of this type should be investigated. In industrial use, there are applications under the current constraints of workpiece shape; however, the problem of picking up non-cubic workpieces should be solved, for example, by setting a virtual cube including such a workpiece. In this study, the suggested method was not compared with other methods because, currently, there are limited methods that can deal with such tiny workpieces in a bin picking problem. The benchmark for bin picking problems with tiny workpieces has not been found, and such problems should be constructed for precise and fair evaluations of performance. Furthermore, an ablation study, that is, a study of the necessity and sufficiency of the suggested method, was not executed. For further applications of the suggested method, the results of this study are invaluable for improving the proposed system.

## 6. Conclusions

An accurate stereo matching method enhanced by a convolutional neural network (CNN) is suggested, and its performance is verified against three-dimensional object detections that emerged in pin picking problems. In previous research, no depth estimation method for tiny cubic workpieces whose edge length is 5 to 10 mm has been suggested. The proposed method, consisting of object detection, instance segmentation, and application of the Harris corner detection algorithm, complements the incomplete depth information obtained via the conventional stereo matching method. A picking plan is generated based on the depth estimation and contact analysis between the robotic arm and the stack of workpieces. The experimental results obtained by investigating two bin picking problems consisting of 10 and 5 mm workpieces show the efficiency and inefficiency of the suggested method. For the former bin picking problem, the suggested method and the picking by the robotic arm were successful; however, for the latter one, object detection and instance segmentation failed for two of the five workpieces. The reason is likely the insufficiency of CNN training. Among the successful recognitions, the errors in depth estimation were acceptable for practical use. In future work, firstly, the reason for the recognition failures of the workpieces in the stack should be pursued, and the suggested method should be applied to various kinds of bin picking problems. Secondly, fair comparisons of the suggested method with other methods and benchmarking of bin picking problems should be executed. Thirdly, the ablation and adequacy of the suggested algorithms must be researched because the current study only verified the feasibility of the suggested method and the bin picking process.

**Author Contributions:** Conceptualization, K.M. and Y.S.; methodology, M.Y.; validation, M.Y.; formal analysis, M.Y.; investigation, K.M. and Y.S.; writing—original draft preparation, M.Y. and Y.S.; writing—review and editing, Y.S.; visualization, Y.S.; supervision, Y.S.; project administration, Y.S. All authors have read and agreed to the published version of the manuscript.

**Funding:** This research received no external funding.

**Data Availability Statement:** Not available.

**Conflicts of Interest:** The authors declare no conflict of interest.

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
