# Peer review of "A Deep Learning-Enhanced Stereo Matching Method and Its Application to Bin Picking Problems Involving Tiny Cubic Workpieces"

_electronics, doi:10.3390/electronics12183978_

Round 1

Reviewer 1 Report

This manuscript proposed a deep learning-enhanced stereo matching method and its application to bin picking problem for tiny cubic workpieces. As far as I am concerned, this work is meaningful and with practical value. To further improve the quality of the manuscript, some issues should be addressed.

1. The main contributions are suggested to be rewritten, so as to better motivate the readers. In addition, some recent research works should be considered in the Introduction section, such as stereo matching-hyperspectral anomaly detection via sparse representation and collaborative representation, stereo matching: hyperspectral anomaly detection via dual dictionaries construction guided by two-stage complementary decision, stereo matching : dual collaborative constraints regularized low-rank and sparse representation via robust dictionaries construction for hyperspectral anomaly detection, and stereo matching: object detection in remote sensing images based on improved bounding box regression and multi-level features fusion.

2. What’s the evaluation metrics used in this manuscript? Please give a detailed information about it in the Experimental Results section.

3. A “Discussion” section should be added to further identify and debate the strengths, but also the weaknesses of the work, and the reference value of this work.

4. The figures may be not clear enough. It will be better if the authors can improve the resolution of the figures, such as Figure 15, Figure 16, etc.

 Minor editing of English language required.

Author Response

Thank you so much for your detailed and valuable review. Please see the attachment for the authors' reply.

Reviewer 2 Report

The manuscript proposes a stereo matching method enhanced by Deep Convolutional Neural Network (D-CNN) based object detection and instance segmentation for the pin picking problem. The novelty of the manuscript lies in enhancing the conventional stereo matching method by employing a D-CNN for a bin picking problem consisting of tiny cubic workpieces. While the proposing method seems promising, it lacks comparison with other methods and lacks an ablation study. The manuscript requires extensive improvements in terms of English language delivery and organization, as some parts of the manuscript are not well explained or organized. Specific recommendations for revising the manuscript are provided below:

1)    The manuscript requires substantial improvements in terms of English language delivery and organization.

2)    The abstract needs refinement in language and delivery.

3)    Please briefly explain the “bin picking problem” when it is first introduced in the abstract section.

4)    Line 29: References should not be referred in the text. Phrases such as "…in [2]… " or "Reference [2] states…" are not acceptable. Please rewrite the sentence.

5)    Line 87 – 88: Please correct the grammar issue in the sentence.

6)    Figure 6 & 7: it is recommended to use different shapes for different purposes. For example, use circles for data and rectangles for processes.

7)    Include comparison methods to benchmark against the proposed method.

8)    Conduct an ablation study and present its findings.

9)    Line 387: Please provide reasoning or evidence to back up the claim that “the reason is likely the insufficiency of CNN training.”

Provided above.

Author Response

(The authors gave the same response as above.)

Reviewer 3 Report

        The paper titled “A Deep Learning-Enhanced Stereo Matching Method and its Application to Bin Picking Problem for Tiny Cubic Workpieces” describes how image recognitions are being used in the appearance examination of products or the autonomous driving of automobiles and mobile robots, and states that the paper deals with the application of D-CNN for solving bin picking problems consisting of tiny cubic workpieces of cube shapes with length less than 10mm..

        Further, its states that bin picking problem is quite common and important in the product manufacturing. This paper deals with the bin picking problem consisting of tiny cubic workpieces which is used in the product manufacturing, the dimension, material, and weight, for example, small chip by a robotic arm as shown in Figure 1, showing future assembly work with tiny workpieces in Fig. 2, all show the importance of bin picking problem and tiny workpieces, Fig. 3 shows images obtained by depth camera and Fig. 4 by 3D scanner and Figure 5 shows examples of bulk stacking and workpieces.

        The process flow of the proposed system is given in Figure 7 with synthesis of depth values and instance segmentation given in Fig. 8, showing depth differences among corners are within the half of the height of workpiece, the layer numbers are obtained precisely. Its accuracy against actual bin picking problems is verified in the experiments using Harris algorithm.

        Object detection and instance segmentation results for bin picking problem with first bin picking problem is shown in Figure 12, here, 10 plastic workpieces whose edge size is 10 [mm] are stacked into three layers. The positions of workpieces in the stack shown in Figure 12 are obtained by object detection with more than 0.99 confidence, and the contour of workpiece is extracted by instance segmentation as shown in Figure 14 (a).

Question 1.     Figure 14 (b) is not referred to in the text as the readers would like to see it in relation to Figure 14(a).

        The commercial software tool, IC3D [22], is used for the conventional stereo matching. The obtained result is shown in Figure 15, here, Figure 15(a) and Figure 15(b) are the images obtained by the left and the right cameras, respectively. Figure 15 (c) is the stereo matching result generated by IC3D.

Author Response

(The authors gave the same response as above.)

Round 2

Reviewer 1 Report

The authors have addressed my concerns.

Reviewer 2 Report

All my comments were addressed.